# Analysis of the Possibility of Using the Plain CFD Model to Simulate Two-Phase Flows in Spatial Systems of Pressure Sewer Networks

**Piotr Siwicki** [1,*], **Marcin Krukowski** [1], **Jan Studziński** [2], **Bartosz Szeląg** [3] and **Rafał Wojciechowski** [4]

1   Faculty of Civil and Environmental Engineering, Warsaw University of Life Sciences SGGW, PL 00-090 Warsaw, Poland; marcin_krukowski@sggw.edu.pl
2   Systems Research Institute PAN, PL 00-090 Warsaw, Poland; Jan.Studzinski@ibspan.waw.pl
3   Faculty of Environmental, Geomatic and Energy Engineering, Kielce University of Technology, PL 25-116 Kielce, Poland; bszelag@tu.kielce.pl
4   EkoWodrol Sp. z o.o., PL 75-209 Koszalin, Poland; rafal.wojciechowski@ekowodrol.eu
*   Correspondence: piotr_siwicki@sggw.edu.pl; Tel.: +48-22-5935293

**Abstract:** The paper analyzes the possibility of using the CFD (Computational Fluid Dynamics) method to predict the amount of sewage remaining in siphons after a full air blast of the pressure sewer system. For this purpose, the results from measurements carried out on a laboratory installation were compared with the results obtained from modelling using a spatial model (3D) and a plain model (2D) of the installation. To determine these models, the structure of the VOF (Volume of Fluid) model was used in the CFD method. The simulation calculations carried out make it possible to state that the use of the plain model with the development of the installation modelled in the plan does not result in significant deterioration of the obtained results. The possibility of using 2D models for modelling pumped sewer systems allows for a significant shortening of the calculation time, which, in practice, results in the possibility of modelling much larger and longer installations than is possible with 3D models.

**Keywords:** pressure sewer installations; siphons; laboratory tests; CFD modeling; volume of fluid (VOF) method

---

## 1. Introduction

In the last few years, there has been a significant development of pressure sewer systems due to the need to adapt designed installations to the difficult topographic conditions in which they operate, such as: a small slope of the terrain; striped buildings; rocky ground; and unfavorable ground conditions with high groundwater levels [1–6]. The main advantage of such installations is the tightness of the pipeline and the resulting lower odor nuisance of pumped wastewater. Moreover, such sewer systems are characterized by a simpler construction than gravity systems and are cheaper to implement [7,8].

However, during the operation of a pressure sewer system, usually the sewage inside the pipeline is crumpled due to the lack of air access, and dangerous toxic volatile substances are released in expansion wells [9]. This means that the pressure sewer system eliminates the release of odors along the length of the pipeline, but leads to a dangerous situation of hydrogen sulfide release at the end of the pipeline. Hydrogen sulfide in small concentrations causes odor discomfort, while in large concentrations, it is poisonous and may even threaten the lives of waterworks workers.

This phenomenon is prevented by shortening the time of sewage retention in the pipeline, the blowing of compressed air, or the addition of chemical reagents to the sewage. The use of blowing

is a very common and radical method, because the rotten wastewater deposited in the pressure pipe is completely removed from it. This method was used as early as 1942 in sewage systems in California [10].

The method, however, causes some problems when introducing wastewater into the pipeline after the blowing, due to the pipeline profile. The pressure pipe is usually not horizontal in length, but has various, sometimes large slopes and elevations due to the land obstacles that it has to overcome. This results in a number of points with very different elevation coordinates in its profile, where siphons and so-called pipe tops are located.

There is usually some volume of sewage remaining in these siphons, while air accumulates at the top of the pipeline, forming so-called air bags. Wastewater remains in the siphons, i.e., in local pipe minima, and air bags in the upper parts of the pipes, i.e., in local pipe maxima, have an adverse effect on the pressure distribution in the pipeline when the wastewater is once again pumped into the pipeline, and consequently on the operation of the pumps in the source well and on the flow of wastewater in the pipe [11].

Air remaining in the upper parts cause hydraulic hammers in the pipeline and lead to a temporary two-phase flow and a turbulent flow with respect to wastewater. In order to limit such situations as much as possible by controlling the pump operation, it is important to correctly recognize the phenomena of sewage leaching from siphons, and air bags leaching from the tops of the pressure pipe. For this purpose, accurate models of the sewer pressure pipes and the processes that take place in them during air blowing and when pumping wastewater after blowing should be developed. This is a contemporary challenge in sanitary engineering, in which various research facilities are involved by carrying out appropriate laboratory tests, e.g., in Spain [12].

The paper deals with the problem by using the CFD (Computational Fluid Dynamics) method to model the pressure pipeline of aerated wastewater. The model was calibrated on the basis of real data obtained from measurement experiments carried out on a laboratory installation made especially for this purpose in the Environmental Engineering Company EkoWodrol in Koszalin (Poland).

The CFD method is used to create numerical models of the examined environmental processes and is very accurate, though its applicability is limited in the case of modelling complex processes, which include two-phase flow phenomena occurring in pressure sewer pipes. The calculation of spatial models of such processes requires very long calculation times, especially when the model takes into account the variable hydraulic and geometric conditions of the installation under investigation. The pipelines of pressure installations have lengths ranging from several hundred meters to several or even several dozen kilometers, and spatial models of such installations require such long calculation times that they are unacceptable in practice.

Therefore, using access to measurements from the laboratory installation, a 3D spatial model was developed and calibrated for it and then, on the basis of this model, 3D and 2D models were developed for a section of the installation containing a siphon. This made it possible to thoroughly examine the process of leaching the liquid from the siphon during blowing of the pipe with compressed air, and thus examining one of the two key and previously mentioned processes taking place in the pressure sewer. The 2D model requires much less work and calculation time than the 3D model. With results obtained for both models, it was possible to assess whether it is possible to replace the spatial model with a plain model while maintaining satisfactory calculation accuracy and, at the same time, checking whether the calculation times obtained with the 2D model are acceptable from the point of view of calculation practice.

Investigating the processes of flow and aeration of wastewater in pressure pipes by means of measurement experiments and mathematical modelling is of cognitive value, but most importantly, it enables the development of algorithms to control these processes in such a way as to eliminate the phenomenon of wastewater crushing in pipes and the formation of toxic or, at best, bothersome hydrogen sulfide evaporations.

At the same time, it should be noted that adequate aeration of a pressure sewerage system leads to an improvement in the quality of wastewater entering the sewage treatment plant, which has a positive impact on the operation of the plant and on the reduction of pollutant concentrations in treated wastewater. The effect of cyclic aeration of wastewater in sewage pipes is a reduction in the amount of excessive sludge generated in the treatment process, a reduction in the degree of internal sewage recirculation and a reduction in the amount of oxygen supplied to the activated sludge chambers. Reducing the degree of sewage recirculation and reducing the amount of oxygen pumped by wastewater blowers into the bioreactor reduces the electricity consumption of energy-intensive plant facilities. On the other hand, the reduction in the amount of excessive sludge produced by the plant makes it possible to plan less material-intensive activated sludge chambers of smaller capacity at the plant design stage.

From the above, it can be seen that the modeling of sewage pressure systems, creating the basis for the next stage of the development of sewage aeration algorithms, is important both for research as well as for economic and environmental reasons, which is an argument for the purposefulness and even the necessity of such research.

## 2. Materials and Methods

Modelling of the sewage pressure pipeline consisted of three stages. In the first stage of the work, a 3D spatial model of a 100 m long laboratory installation was made and its hydraulic calibration was carried out under conditions of liquid flowing in the installation without air. To develop the model of the installation, the CFD method using the VOF (Volume of Fluid) model was applied [13,14].

In the second stage of research, 3D and 2D models were formulated for a fictitious installation with a siphon, also 100 m long, into which compressed air was entered to test the process of leaching the liquid from the siphon. The main aim of this study was to compare the calculation times obtained for the spatial and plain model.

Finally, in the third stage of the research, a section of the pipe containing a siphon was separated from the laboratory installation in order to enable the testing of the phenomenon of leaching the liquid from the siphon at different speeds of the air intake on the basis of this section. For the extracted section of the installation, with a length of 19.83 m, 2D and 3D models were again determined for different air velocities and the modelling results obtained with these models were compared for both accuracy and calculation time.

### 2.1. Laboratory Installation

The laboratory installation was completed in EkoWodrol's laboratory in Koszalin as a part of a research project "Development of an innovative device for flushing and aerating the sewer pressure pipeline with compressed air limiting the processes of wastewater crushing", financed by the Polish National Research and Development Centre (NCBR) (Figure 1).

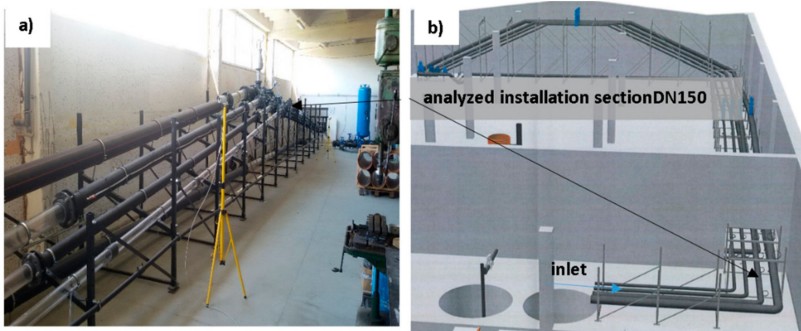

**Figure 1.** View of laboratory installation for testing processes in pressure sewerage pipelines; (**a**) general view (left), (**b**) diagram of the route and profile of the pressure pipeline with indication of the tested section with a siphon (right).

The model of the whole installation was made for a DN150 diameter and a 100 m long pressure pipeline. In the case of modelling a section of the installation with a siphon, the pressure pipeline included in the model has a length of L = 19.83 m and the slope of its ascending segment coming out of the siphon is 6° (Figures 1 and 2).

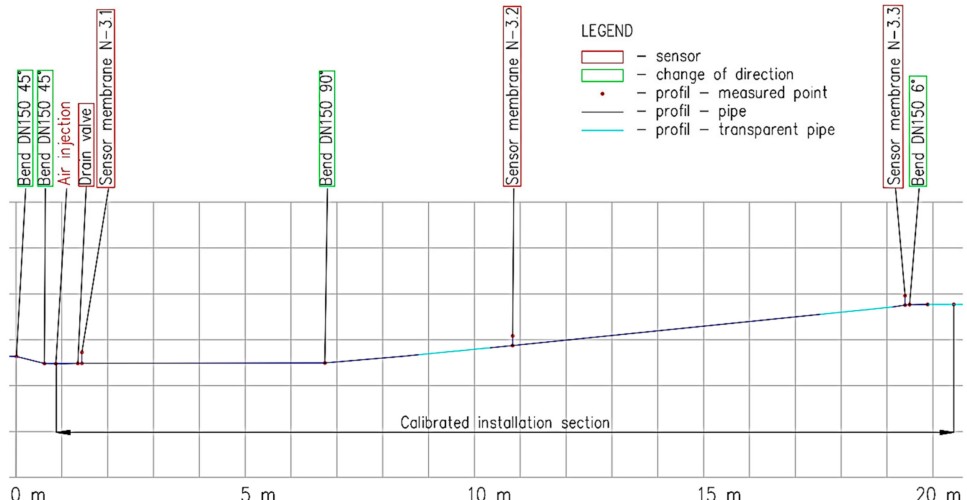

**Figure 2.** Profile of the analyzed installation section with a siphon 19.83 m in length; the liquid flow is from left to right.

The measurement experiments carried out on the laboratory installation were as follows:

- The first stage of the tests consisted in filling the installation with water and removing air from it using vent valves.
- Then the stage of calibration of the 3D model of the whole installation was followed by adjusting the roughness coefficient values of the pipe so that the pressure values calculated by the model coincided with the pressure values recorded in the measurement points located on the installation (Figure 3).
- During these tests, water was pumped into the pipeline at a speed of 0.7 m/s.

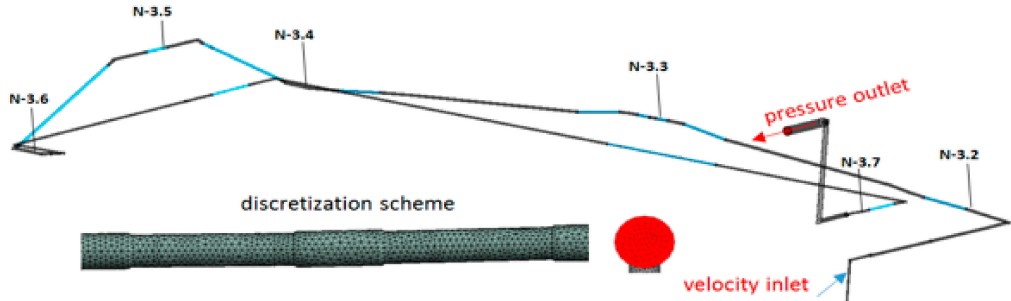

**Figure 3.** Geometry and method of discretization of the 3D model of the entire 100 m long laboratory installation.

Next, after switching off the pump, it was waited until the pressure in the pipeline had stabilized, and, when this happened, the next stage of testing was started with the introduction of air into the pipe.

- The air was introduced at point D-1.3 (Figure 2) and its feeding was completed after 30 s from the moment when air got through the highest point in the siphon; the volume of water remaining in the siphon after such a blow-out was measured by draining water from the trap with a drain valve (Figure 2).

- The air was introduced into the pipe at four speeds used in the operating practice of water supply companies during the blow-out of the pressure systems: v = 0.8; 1.0; 1.2; 1.4 m/s.
- The aim of this stage was to test the process of leaching liquids from the siphons during the blow-out of the pressure pipelines.

The use of water in research instead of sewage was caused by environmental restrictions.

*2.2. Numerical Model*

The CFD method (computational fluid dynamics) and the Fluent program [15] were used to numerically simulate laboratory installation models. To solve the model equations, the Finite Element Method has been applied [16,17]. It is an approximate method to solve partial differential equations describing a control volume of fluid flowing in a modelled pipe. The differential equations in the CFD method are, in general, a mathematical description of the fundamental laws of mass, momentum and energy preservation defining the flow of fluids [18]. Regardless of the type of fluid movement (laminar or turbulent), the following flow continuity equation and momentum equation should always be solved [19–21]:

Flow continuity equation:

$$\frac{\partial \rho}{\partial t} + \frac{\partial u_i}{\partial x_i} = 0 \tag{1}$$

where: $t$—time [s], $u$: fluid velocity [m/s], $\rho$: fluid density [kg/m$^3$] and $u_i$ is the velocity component in the $x_i$-direction (i.e., $u$ and $v$ in the horizontal $x$ and the vertical $y$ directions, respectively).

Momentum Equation (Navier–Stokes) describing the changes in the speed field:

$$\frac{\partial(\rho u_i)}{\partial t} + \frac{\partial}{\partial x_j}(\rho u_i u_j) = \frac{\partial}{\partial x_j}\left[\mu\left(\frac{\partial u_i}{\partial x_j} + \frac{\partial u_j}{\partial x_i}\right)\right] - \frac{\partial p}{\partial x_i} + \frac{\partial \tau_{ij}}{\partial x_j} + F \tag{2}$$

where $p$: hydrostatic pressure, $\mu$: dynamic viscosity, $\tau_{ij}$: strain tensor, $\rho g$: mass forces, $F$: surface forces. The components ($p, u$) in the equation are averaged.

The strain tensor is defined by the following relationship:

$$\tau_{ij} = -\rho\overline{u_i' u_j'} = \mu_t\left(\frac{\partial u_i}{\partial x_j} + \frac{\partial u_j}{\partial x_i}\right) - \frac{2}{3}\rho\delta_{ij}k \tag{3}$$

where $\rho\overline{u_i' u_j'}$: called the Reynolds stresses [4], $\mu_t$: eddy viscosity and $k = (u_i u_j)/2$ is the turbulent kinetic energy per unit mass. There is also a kinematic turbulent or eddy viscosity denoted by $v_t = \mu_t/\rho$, with dimensions m$^2$/s; $\delta_{ij}$ is the Kronecker delta ($\delta_{ij} = 1$ if $i = j$ and $\delta_{ij} = 0$ if $i \neq j$).

The averaged Navier–Stokes Equations [22] for the stationary flow of liquid can be written for a coordinate as follows:

$$\frac{\partial u_j}{\partial x_i} = 0 \tag{4}$$

$$\frac{\partial}{\partial x_j}(\rho u_i u_j) = -\frac{\partial p}{\partial x_i} + \frac{\partial}{\partial x_j}\left[(\mu + \mu_t)\left(\frac{\partial u_i}{\partial x_j} + \frac{\partial u_j}{\partial x_i}\right)\right] - \frac{2}{3}\frac{\partial}{\partial x_i}(\rho k) \tag{5}$$

where, $u_i$: averaged velocity vectors, $\mu_t$: turbulent viscosity, $k$: kinetic turbulent energy, $i,j = 1,2,3$.

Equation (5), by taking into account the turbulent viscosity $\mu_t$, is a description of the transient flow occurring during the mixing of fluids in a two-phase flow. When dealing with a turbulent flow, the transport Equation must also be included in its description. In such a case, the $k$-$\varepsilon$ standard model [15] is most commonly used. The models belonging to the $k$-$\varepsilon$ group are semi-empirical models described by Equations (6) and (7), where one Equation describes the transport of kinetic turbulent energy $k$ [23] and the other describes the dissipation of energy $\varepsilon$. These Equations have been derived

assuming the isotropy of turbulence, whereby this assumption requires the definition of the following additional dependencies for wall flows.

The following transport equation was used in our calculations:

$$\frac{\partial}{\partial x_j}\left(\rho k u_j\right) = \frac{\partial}{x_j}\left[\left(\mu + \mu_t\right)\frac{\partial k}{\partial x_j}\right] + \mu_t\left(\frac{\partial u_i}{\partial x_j} + \frac{\partial u_j}{\partial x_i}\right)\frac{\partial u_i}{\partial x_j} - \rho\varepsilon \tag{6}$$

$$\frac{\partial}{\partial x_j}\left(\rho\varepsilon u_j\right) = \frac{\partial}{x_j}\left[\left(\mu + \mu_t\right)\frac{\partial k}{\partial x_j}\right] + C_{\varepsilon 1}\mu_t\left(\frac{\partial u_i}{\partial x_j} + \frac{\partial u_j}{\partial x_i}\right)\frac{\partial u_i}{\partial x_j}\frac{\varepsilon}{k} - C_{\varepsilon 2}\rho\frac{\varepsilon^2}{k} \tag{7}$$

In Equations (6) and (7), the parameters $k$ and $\varepsilon$ are linked by the following phenomenological relationship, based on the observation of real flows:

$$\varepsilon = \frac{k^{3/2}}{L} \tag{8}$$

where $\varepsilon$: dissipation of kinetic turbulent energy, L: length scale, which may be expressed by the hydraulic radius L = 0.07D [15].

Relation (8) results from the Kolmogorov model [24], which assumes that the kinetic energy of turbulence k is transferred from large scale whirlpools to small scale whirlpools where it is dissipated.

The unknowns in the transport Equation are calculated based on the following relationship described in [25]:

$$\mu_t = C_\mu\rho k^{1/2}L = C_\mu\rho\frac{k^2}{\varepsilon} \tag{9}$$

Universal constant values for incompressible liquids $C\mu$, $C\varepsilon_1$ and $C\varepsilon_2$ in Equations (7) and (9) are determined experimentally. The following values of these coefficients were determined in our calculations: $C\mu$ = 0.009, $C\varepsilon_1$ = 1.44 and $C\varepsilon_2$ = 1.92 [15].

### 2.2.1. Multi-Phase Flows

CFD methods enable the modelling of many types of multiphase flows, in which the following combinations of flowing fluids can generally be distinguished: gas-liquid, liquid-liquid, gas-solid, liquid-solid for two-phase flows and gas-liquid-solid for three-phase flows. Two versions of the CFD method are currently used to numerically solve multiphase flow Equations: The Euler–Lagrange method and the Euler–Euler method [15,17,26,27].

Our calculations use the Euler–Euler method, which assumes that all the phases considered are treated as mutually interpenetrating continuums. This method can use different descriptions of created models: the Volume of Fluid model (VOF), the Mixture model and the Eulerian model. The VOF model was used in the calculations. This model is one of many commonly used proposals for the reconstruction of the interfacial surface based on the balancing of a certain additionally accepted quantities: phase content fraction $\alpha_q$ [15]. The $\alpha_q$ function is a scalar function assuming values from 0 to 1, depending on the volume amount of a given phase f in the control fluid volume.

The tracking of the interface between the phases is accomplished by the solution of a continuity Equation [5] for the volume fraction of one (or more) of the phases. For the qth phase (10), this Equation has the following form:

$$\frac{1}{\rho_q}\left[\frac{\partial}{\partial t}\left(\alpha_q\rho_q\right) + \nabla\cdot\left(\alpha_q\rho_q u_q\right) = S_{\alpha q} + \sum_{p=1}^{n}\left(m_{pq} - m_{qp}\right)\right] \tag{10}$$

where $m_{qp}$ is the mass transfer from phase $q$ to phase $p$ and $m_{pq}$ is the mass transfer from phase $p$ to phase $q$. By default, the source term on the right-hand side of Equation (10), $S_{\alpha q}$, is to specify a constant or user-defined mass source for each phase.

The volume fraction Equation will not be solved for the primary phase; the primary-phase volume fraction will be computed based on the following constraint (11):

$$\sum_{q=1}^{n} \alpha_q = 1 \tag{11}$$

Based on the knowledge of the solution of Equation (11), i.e., the distribution of values and gradient $\alpha_q$ in space, the actual surface defined as a functional layer with $\alpha_q$ value equal to $\frac{1}{2}$ is reconstructed, assuming arbitrarily that this value best approximates the position of the interphase boundary.

From among a number of available algorithms used to reconstruct this surface, an algorithm based on a linear interpolation of $\alpha$ values, called the geometric reconstruction scheme, was selected for calculation. In the first stage of calculations, it determines the positions of the layered points on the edges of finite elements by means of linear interpolation. The second stage of the calculation consists of determining the surface based on the marching cubes algorithm.

This algorithm, widely used in the field of three-dimensional space graphics, allows for the quick and effective determination of the surface, with accuracy limited to the size of a single finite element. The resulting geometric structure, consisting of triangles, approximates the actual geometry of the interfacial surface.

The modelling calculations were carried out under steady state conditions, thus obtaining approximate solutions. This way the solutions of the above equations were obtained faster than using transient (transident) modelling. However, the adoption of this modelling approach did not allow for the use of real time steps in the calculations, forcing only iterative steps to be taken into account, which meant that it was not possible to trace the changes in the process of the leaching of liquids from the siphon on the numerical model in real time.

### 2.2.2. Boundary Condition

Defining boundary conditions is one of the most important tasks when performing CFD analyses. The following boundary conditions were used in the models under study:

- the inlet to the model was defined as a function velocity inlet, in which a constant value of velocity (of liquid or gas in case of blowing) was given, without giving the velocity profile in the cross-section of the modelled pipeline; the volume fraction for the primary phase $q$ (air) was equal to 0 for water or 1 for air; the turbulence parameters length scale and hydraulic diameter were adopted as the specification of transported turbulence quantities;
- on the wall of the pipeline, adopting the standard wall roughness models [15], the concept of the boundary layer was used, assuming the change of velocity profile in turbulent flow near the wall,
- the values of viscosity and density of liquids were assumed for liquids (water), and for gas (air);
- the model outlet was defined as a pressure outlet function; the turbulence parameters length scale and hydraulic diameter were adopted as the specification of transported turbulence quantities.

### 2.3. Spatial Model (3D) of the Installation

On the basis of the technical documentation of the 100 m long laboratory installation, its spatial 3D model was developed (Figure 3). For this model, a spatial calculation grid was defined, consisting of 1,888,302 knots and 1,521,879 elements. An average grid quality index of 0.83 was obtained, while the maximum value of this indicator is 1. To carry out calculations with such a large number of elements, 16 processors with a 2.4 GHz clock frequency were used.

The boundary condition at the point of pumping water into the installation (inlet) was defined in the model as a velocity-inlet function of the FLUENT program, and, at this point, the value of water flow velocity was defined. The boundary condition at the point of water outflow from the installation

(outlet) was defined in the model as a pressure-outlet, meaning free flow. Using the function wall, the pipe walls were defined.

During the calibration of the model of the sewer pipeline, a value of the coefficient of pipeline roughness equal to 0.0001 m was determined. The calculated pressure values were recorded at the same points of the model where the pressure sensors on the laboratory installation are located (Figure 3).

### 2.4. Spatial Siphon Model (3D)

On the basis of the profile of the entire laboratory installation (Figure 2), a 19.83 m long section of the installation containing a siphon has been extracted, and a 3D spatial model has been developed for this pipeline fragment.

The modelled section of the installation has a diameter of DN150, with the slope of the ascending pipe being 6° (Figure 4). For the model being created, a spatial calculation grid has been defined, consisting of 2,327,135 knots and 1,583,009 elements. An average grid quality index of 0.96 was obtained.

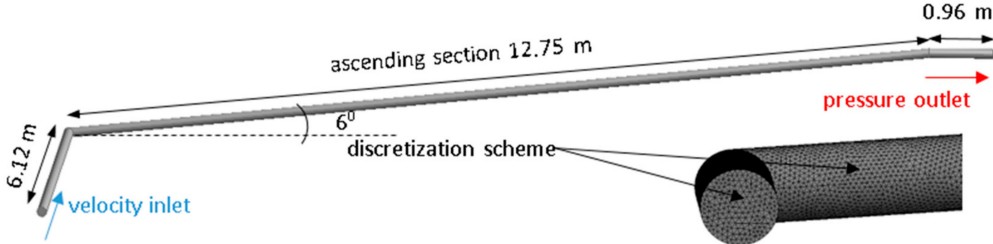

**Figure 4.** Geometry and method of discretization of the 3D model of the 19.83 m long installation section containing the siphon.

The boundary conditions in the model were defined by assuming the function velocity inlet, where the air was introduced with different velocity values assumed, while the model outlet was defined as the function pressure outlet. For pipeline walls the function wall was assumed. The value of roughness coefficient determined for the pipe is equal to $k = 0.0001$ m.

### 2.5. Plain Siphon Model (2D)

By creating a 2D model for the pipeline with a siphon, the relevant spatial fragment of the installation is presented on the plane while respecting the length of individual pipeline sections (Figure 5). On the basis of the flat profile of the pipeline obtained in this way, a two-dimensional model was developed, whose calculation grid consists of 27,236 knots and 25,587 elements. An average grid quality index of 0.98 was obtained.

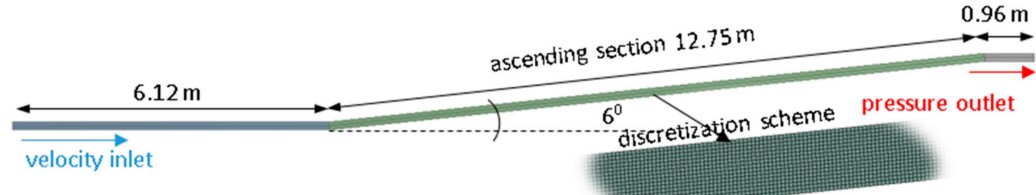

**Figure 5.** Geometry and method of discretization of the 2D model of the 19.83 m long installation section containing the siphon.

Similar to the 3D model, the length of the modelled pipeline is 19.83 m, the pipe has a diameter of DN150, the slope of the ascending section is 60 and the pipeline roughness coefficient is $k = 0.0001$ m. The functions velocity inlet, pressure outlet and wall are assumed for the inlet, outlet and wall of the modelled pipeline respectively.

## 3. Results

In the first phase of research, a 3D model of the laboratory installation was developed and calibrated on the basis of pressure measurements from six measurement points installed on the installation (Figure 3). The results from the model simulation (Figure 6) were compared with the measurement results leading to their convergence by choosing the value of the roughness coefficient of the pressure pipe. Pressure values from sensors N-3.2, N-3.3, N-3.4, N-3.5, N-3.6 and N-3.7, located virtually in the model and physically on the pipeline, were analyzed. For comparison of the calculated and measured pressures, the average values determined over a water flow duration of 103 s were taken into account. The average water flow value determined from the flow meter installed in the source well of the installation, equal to 18.52 m$^3$/h, was also considered in the calibration calculations.

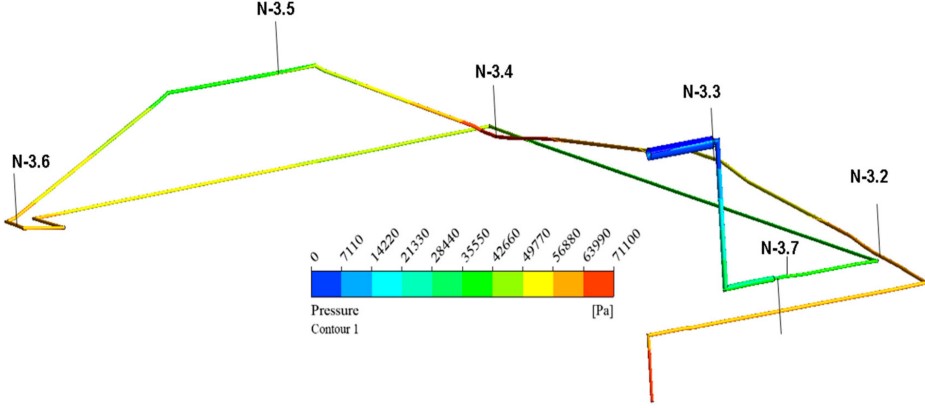

**Figure 6.** Pressure distribution obtained during calibration of the CFD model of the entire 100 m long laboratory installation.

During the calibration of the model, a high correlation between the results of the calculation and the measurements was obtained, assessed on the basis of the correlation coefficient. For the calibrated model, a value of this coefficient equal to R$^2$ = 0.99 was obtained (Figure 7). Only for the sensor N-3.7, placed close to the outlet of the installation, was greatest difference in pressure values measured and calculated at the level of 2% obtained, which results from the unsteady flow conditions at this point of the pipeline. This is evidenced by the oscillations in the readings of the momentary flow values from the flow meter installed at the outlet of the installation.

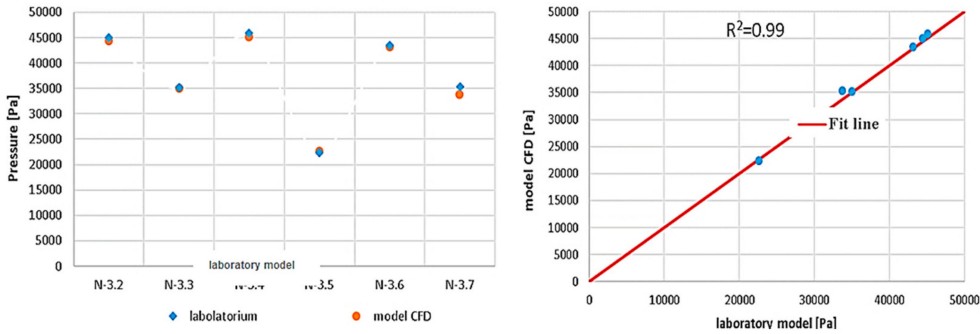

**Figure 7.** Comparison of pressure values obtained from the CFD model with the results obtained from the measurements.

The time of a single simulation of the 3D model for the 100 m long installation during the model calibration was 48 h. A PC with 16 processors and with a 2.4 GHz clock frequency was used to carry out the calculations and water was pumped into the pipeline at a velocity of 0.7 m/s.

In the next phase of research, 3D and 2D models were used to carry out simulations for a fictitious 100 m long pipe with a siphon, including wastewater aeration, i.e., a two-phase flow in the pressure

pipe was already modeled. In this study, the process of leaching the liquid from the siphon of the pipeline during its blowing with compressed air, pumped into the pipe at a velocity of 0.8 m/s was analyzed. Modelling calculations were carried out until the moment when the injected air reached the outlet of the pipeline. The purpose of these calculations was to check what impact the replacement of a complex 3D model with a simpler 2D model has on reducing the calculation time. It turned out that this impact is very high because the simulation of two-phase flow for the tested pipeline using the 3D model required a calculation time of 168 h, while for the 2D model this time was 24 h, which was seven times shorter. It can be noted that the time of two-phase flow modeling has significantly increased compared to single-phase flow modeling: a single simulation of a 3D model for a 100 m long laboratory installation for single-phase flow was 48 h, while for a 100 m long pipeline, but for two-phase flow, it was as much as 168 h.

In the last phase of research, using 3D and 2D models, simulation calculations were performed for a selected section of a 19.83 m long laboratory installation, analyzing, as before, the process of leaching the liquid from the siphon of the pipeline during its blowing with compressed air. This time, a two-phase flow in the pipe was once again modeled. The aim of this phase of research was twofold. On the one hand, it aimed to determine the influence that the process of leaching the liquid from the installation siphon has on the velocity of air entering the pipeline, with the second equally important aim of the research being the assessment of the impact of replacing the 3D model with a 2D model on the accuracy of calculations. Measurement experiments carried out on the laboratory installation and calculation experiments carried out with the use of the models consisted in pressing air into the pipeline at different velocities of 0.8, 1.0, 1.2 and 1.4 m/s and analyzing the amount of water remaining in the siphon. The process of leaching water from the siphon was carried out on the installation and simulated with the use of models until the entered air reached the outlet section of the tested pipe. At this stage, the simulation was completed and the amount of water remaining was estimated. This means that, as before, the experiment was stopped after the air passed through the analyzed syphon section.

To determine the amount of water remaining in the siphon in the case of the laboratory installation and the 3D model, a dimensionless factor was defined, expressing the ratio of the volume of the pipe occupied by the water to the geometric volume of the analyzed section of the installation.

In the case of the 2D model, a coefficient has been defined expressing the ratio of the field occupied by water to the field of the analyzed section of the installation.

The results of the measurement experiments and calculations performed have been presented in Table 1.

**Table 1.** Values of the residual water volume in the siphon obtained from the measurements and modelling calculations.

| Velocity (m/s) | Amount of Water Remaining (l) | Water Volume Fraction (-) |
|---|---|---|
| | Laboratory test | |
| 0.80 | 145.60 | 0.42 |
| 1.00 | 132.30 | 0.38 |
| 1.20 | 118.65 | 0.34 |
| 1.40 | 102.55 | 0.29 |
| | CFD model 3D | |
| 0.80 | 153.30 | 0.44 |
| 1.00 | 129.50 | 0.37 |
| 1.20 | 95.20 | 0.27 |
| 1.40 | 88.20 | 0.25 |
| | CFD model 2D | |
| 0.80 | - | 0.45 |
| 1.00 | - | 0.35 |
| 1.20 | - | 0.27 |
| 1.40 | - | 0.22 |

Calculations carried out with the help of 3D and 2D numerical models allowed to observe the variability of the water volume in the siphon during the pumping of air into the pipe (Figure 8). The results of the calculations show that the process of leaching the liquid from the siphon of the pressure pipe simulated by means of both models is very similar, which means that the application of 2D modelling generally gives a similar calculation accuracy to that of 3D modelling, at least for the tested two-phase flow in the pressure pipeline.

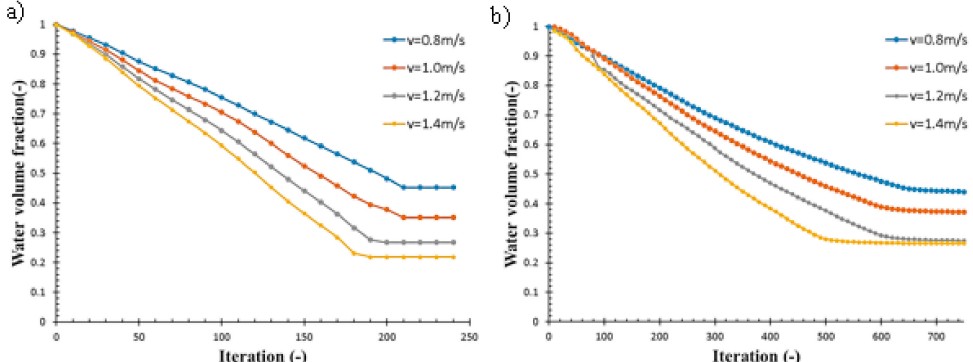

**Figure 8.** Variability of water volume remaining in the installation during the air blow-out obtained from calculations carried out using: (**a**) 2D model, (**b**) 3D model.

The comparison of the calculation results obtained for both models with the measurements is shown in Figure 9. Both the CFD 3D spatial model and the 2D plain model overstate the amount of water remaining in the installation at the assumed maximum blowing speed of 1.4 m/s. For the remaining blowing speeds, the determined volume of water remaining in the siphon is slightly lower than in reality. Despite some discrepancies between the measurement results and the results obtained from the numerical models of the installation, high values of correlation coefficients were obtained for all analyzed cases.

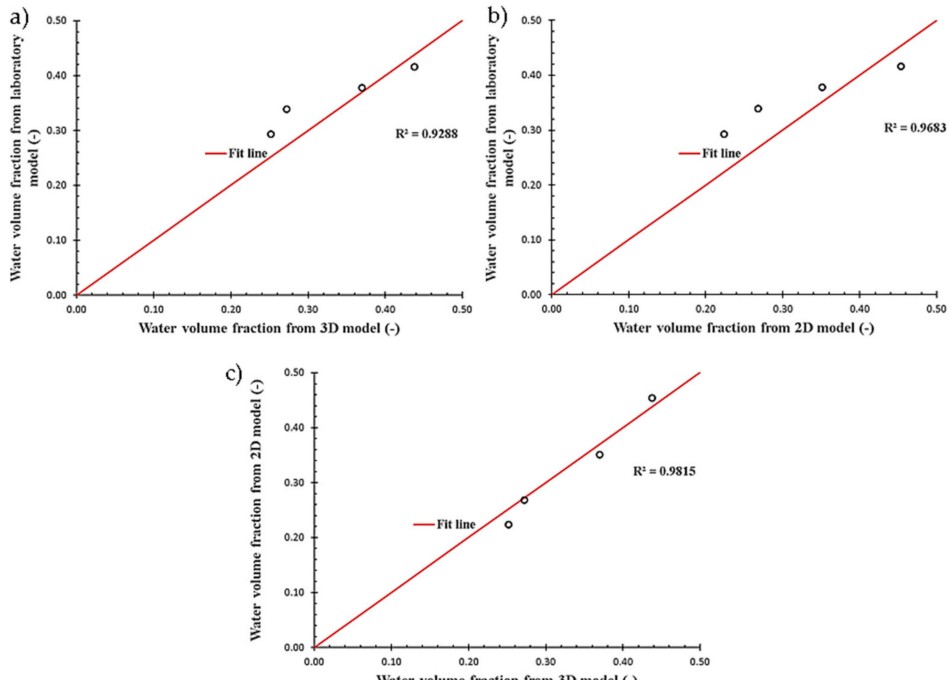

**Figure 9.** Comparison of modelling results with measurement results: (**a**) 3D model vs. measurements, (**b**) 2D model vs. measurements, (**c**) 3D model vs. 2D model.

The correlation between the measurements from the installation and the results of the 3D model was $R^2 = 0.9288$, with that for the 2D model at $R^2 = 0.9683$; the highest correlation coefficient was obtained between the results of the calculations of both models, equal to $R^2 = 0.9815$.

On the basis of the calculations carried out, it can be concluded that, in the analyzed flow and blowing conditions, the modelling of sewage pressure pipes with the CFD method is justified. The obtained results of calculations concerning the leaching of sewage from the installation siphons during blowing with compressed air are very similar to the results obtained from measurements on real installations.

Moreover, it is justified and advisable to use 2D plain models in such cases, seeing as how the results obtained with their use only slightly deviate from the results obtained with 3D spatial models.

At the same time, a very important argument in favor of using 2D modeling to simulate two-phase flows in pressure pipelines is the significant reduction in calculation time compared to 3D modelling. When modelling an isolated 19.83 m long section of the installation with an air velocity of 0.8 m/s, calculation times of 1 h for the 2D model and 7 h for the 3D model were obtained. A calculation time of 1 h is acceptable in calculation practice. At the same time, the obtained results indicate that it is advisable and justified, for the sake of time saving, not only to use 2D for the simulation of two-phase flows in pressure pipes, but also to model selected characteristic flow processes for only selected parts of the pipelines as opposed to the entire sewage system.

## 4. Discussion

In order to improve the quality of wastewater in pressure sewer networks, it is common practice to use compressed air blowing to break the biological film from the pipeline walls and to aerate the wastewater. In the case of full blowing, sewage stoppers are formed at the points of change of the network route (siphons), from which the sewage is not completely leached. The amount of sewage remaining in the installation generates additional pressure in the pipeline and its amount affects pump operation parameters when the sewage is once again pumped into the installation.

High convergence of the results of measurements conducted on the laboratory installation (physical model) with the results obtained by means of the numerical models indicate the usefulness of the CFD method in determining the amount of sewage remaining in siphons after a full blow of the installation. The use of the VOF model for 3D and plain 2D spatial modelling gives results similar to those obtained from the laboratory installation.

The use of a flat model (2D) with the installation development in the plan does not significantly affect the results obtained from the spatial model (3D). At the same time, the possibility of using a 2D model for modelling pressure sewage systems allows for the significant reduction of calculation time and, as a result, enables the modelling of much larger and longer sections of installations than is practically possible with a 3D model.

At the same time, it is important to stress the importance and advisability of conducting research related to modelling the flow and aeration of wastewater in pressure pipelines. Such research is not only scientific in nature, but also very practical, as it enables the development of algorithms for controlling the aeration of the pressure canalization in such a way as to eliminate the crushing of sewage and the formation of bad odors that are potentially harmful to health. At the same time, appropriate aeration of wastewater in pressure pipes has a positive impact on the operation of wastewater treatment plants, minimizing their electricity consumption and having a positive impact on the quality of treatment by supporting the process of reducing pollution in wastewater, which has important ecological significance.

**Author Contributions:** Conceptualization, P.S., methodology, P.S., M.K., J.S., B.S.; formal analysis, P.S., M.K., J.S., B.S.; writing—original draft preparation P.S., M.K., J.S., B.S.; writing—review and editing, P.S., M.K., J.S., B.S. and R.W.; visualization, P.S., M.K., J.S., B.S. and R.W.; supervision, P.S., M.K., J.S., B.S. and R.W.; All authors have read and agreed to the published version of the manuscript.

**Funding:** Measurement data, on the basis of which modelling calculations were made, were obtained during experiments carried out at EkoWodrol in Koszalin/Poland under the research project No. 01/11/2016/NCBR entitled: "Development of an innovative device for flushing and aeration of wastewater pressure pipeline with compressed air, limiting the processes of wastewater crushing", co-financed by the Polish National Centre for Research and Development.

**Conflicts of Interest:** The authors declare no conflict of interest.

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
