# Peer review of "Analysis of the Possibility of Using the Plain CFD Model to Simulate Two-Phase Flows in Spatial Systems of Pressure Sewer Networks"

_water, doi:10.3390/w12061779_

Round 1

Reviewer 1 Report

The paper describes numerical experiments using CFD to model two-phase flow in pipes representing components in a sewer network. A commercial code, Ansys-Fluent, is used in both 2D and 3D simulations. An initial comparison is made to a laboratory experiment in order to calibrate the pipeline roughness used in the remaining simulations. The main objective of the paper is to show that 2D simulations give acceptable accuracy, as compared to 3D simulations and laboratory results, with a great saving in computer time.

The paper is well-written, and the results should be useful to designers of sewer networks. Publication is recommended, following some small corrections or clarifications, as follows.

The graphs in Fig. 9 all seem to have their axes transposed, by comparison with the results in Fig. 8 and Table 1. The axes corresponding to laboratory and simulated results in the upper two graphs are interchanged, and the 2D and 3D axes in the lower graph are also interchanged.

Does “iteration” correspond to time in Fig. 8?

In lines 137 and 265, should “60” be replaced by 6 degrees?

There is a weird symbol in line 178 denoting turbulent viscosity.

In several instances, the word “labolatory” is used instead of “laboratory”.

Author Response

Drogi recenzencie

DziÄ™kujemy za cenne komentarze i wskazówki. Nasze odpowiedzi w załączniku

Z poważaniem,

Autorzy publikacji

Reviewer 2 Report

The first problem with this article is that it does not meet what its title "Analysis of the possibility of using the plain CFD model ..." promises, not even remotely! This starts with line 67 where the authors mention "*THE* CFD model" which is ridiculous: CFD is a collective term embracing a large number of very very different models and analytical, semi-analytical and numerical methods, finite differences, VOF, FEM, boundary elements, spectral methods, ... and a serious analysis of this matter would start with an overview over all of this and would consider more than one method, and, of course, also contributions from other authors dealing with multiphase flows. What in fact is presented here is just a comparison of the results from a commercial software package with experiments. This very poor contribution lives not up to the promise of its title!

The second problem is the transparency of the experimental work:

  • The project "Development of an innovative device for flushing and aerating the sewer pressure pipeline with compressed air limiting the processes of wastewater crushing", where the data has been taken from has to be listed under "Funding" at the end of the article, including the grant number! I did not find anything about this project be literature research!
  • Are the experimental results already published, e.g. in a technical report about the project? In this case this needs to be cited!
  • How are the authors involved in this project? 

The third problem is the formulation of the equations: first, on about two pages textbook knowledge is provided about Navier-Stokes equations, continuity equation and turbulent standard models, known by everyone, but nothing is provided about boundary conditions, which are always relevant for problems with PDE, but in particular for multiphase problems! Also, the following statements on the multiphase problem beginning with line 200 are very vague, featureless, unclear and without formulas that would be important in this context since this is really the most tricky part of the entire modeling going beyond standard fluid mechanics.

The fourth but equally serious problem is the sloppy and partly erroneous state of the manuscript. Here are some examples for it:

  • line 72: "The CFD method is used to create physical models ..." - this sounds rather odd, since usually vice versa a numerical method is implemented for a given physical model. 
  • Figure 2 is of extremely poor quality. The other figures could also be improved.
  • line 176: "flow of an incompressible liquid" is nonsense! Incompressibility is not a matter of the liquid alone! Incompressibility is a matter of the Mach number, the ratio of the characteristic velocity to the speed of sound!
  • Some equations are presented within Nabla calculus, but not really accurate (missing dot between nabla and vector indicating a divergence and no hint that 'vv' denotes a tensor product), some others are given in tensor notation but without mentioning Einstein's summation convention. This may confuse the reader. Also the typesetting changes sometimes: in Eq. (5) the velocity is denoted by an italic u, in Eq. (6) with a roman one.
  • Line 196 "... and the following two differential equations:" There are no equations! They seem to be forgotten.

There may be many other points but it makes no sense to find them all.

My overall impression is that this manuscript was written in a hurry. It is far away from being considered as a scientific paper.

Author Response

Dear Reviewer

Thank you for your valuable comments and tips. Our answers in the attachment

Kind regards,

Authors of the publication

Round 2

Reviewer 1 Report

My suggested changes have been made. I recommend publication.

Author Response

Thank you for your valuable comments when editing the manuscript.

Reviewer 2 Report

Despite some improvements there are still some points that need to be addressed:

1) The authors seem to be fully unfamiliar with the nabla calculus: In eq. (3) they use nabla with a dot for two completely different operations: (1) a Jacobean and (ii) a divergence. Normally a nabla with an 'otimes' symbol denotes a Jacobean. And - what is completely confusing - they use just the nabla without anything for a divergence in eq. (1) ... plus they use in eq. (2) they use the pure nabla once for a gradient (nabla p) and for the divergence of a tensor (nabla tau) which are again two completely different things. This his fully hopeless!!!! At least, the following equations with index notation are correct, so I urgently recommend to re-write equations (1) - (3) in index notation! There is really no need to use nabla calculus for three equations and index notation for all others!!

2) The authors state in their reply that their "calculations [...] were not financed by any research grant or project", but they publish the experiments from the company alongside to their numerical results for comparison. And the experiments are performed with financial public support. I am convinced that this 'indirect support' should be transparently explained at the end of the article! But this is up to the journal's editor to clarify this ...  

Author Response

Thank you for your valuable comments. Proposed amendment to the attachment
